# *HER2* and *BARD1* Polymorphisms in Early HER2-Positive Breast Cancer Patients: Relationship with Response to Neoadjuvant Anti-HER2 Treatment

**DOI:** 10.3390/cancers15030763

**Published:** 2023-01-26

**Authors:** Apolonia Novillo, María Gaibar, Alicia Romero-Lorca, Diego Malón, Beatriz Antón, Amalia Moreno, Ana Fernández-Santander

**Affiliations:** 1Medicine Faculty, Cellular Biology Section Department, Complutense University, 28040 Madrid, Spain; 2Human Genetic Variability Group, Hospital La Paz Institute for Health Research–IdiPAZ (La Paz University Hospital–Universidad Autónoma de Madrid–Getafe University Hospital–Universidad Europea de Madrid), 28046 Madrid, Spain; 3Faculty HM Hospitales de Ciencias de la Salud de la UCJC, Universidad Camilo Jose Cela, 28707 Madrid, Spain; 4Biomedical and Health Sciences Faculty, Department of Medicine, Universidad Europea de Madrid, 28670 Madrid, Spain; 5Department of Oncology, University Hospital of Fuenlabrada, 28942 Madrid, Spain; 6Department of Oncology, University Hospital of 12 de Octubre, 28041 Madrid, Spain

**Keywords:** HER2-positive breast cancer, anti-HER2 treatment, *HER2* gene, SNPs, pathological complete response, Miller–Payne grading

## Abstract

**Simple Summary:**

Neoadjuvant treatment with anti-HER2 drugs such as trastuzumab or pertuzumab improves outcomes in patients with HER2-positive breast cancer. However, resistance to this treatment in some patients determines a need to identify genetic biomarkers able to predict patient responses and optimize personalized treatments. In this work, two different SNPs (rs1058808 HER2Ala1170Pro and rs2070096 BARD1Thr351=) are proposed as potential biomarkers of a good response to anti-HER2 treatment in patients with early HER2-positive breast cancer.

**Abstract:**

The addition to chemotherapy of anti-HER2 drugs such as trastuzumab or pertuzumab has improved outcomes in HER2-positive breast cancer patients. However, resistance to these drugs in some patients remains a major concern. This study examines the possible association between the response to neoadjuvant anti-HER2 treatment in breast cancer patients and the presence of 28 SNPs in 17 genes involved in different cell processes (*PON1*, *CAT*, *GSTP1*, *FCGR3*, *ATM*, *PIK3CA*, *HER3*, *BARD1*, *LDB2*, *BRINP1*, chr6 intergenic region, *RAB22A*, *TRPC6*, *LINC01060*, *EGFR*, *ABCB1*, and *HER2*). Tumor samples from 50 women with early breast cancer were genotyped using the iPlex^®^Gold chemistry and MassARRAY platform, and patients were classified as good responders (Miller–Payne tumor grades 4–5) and poor responders (Miller–Payne tumor grades 1–3), as assessed upon surgery after 6 months of treatment. Proportions of patients with the *HER2Ala1170Pro* (rs1058808) SNP double mutation were higher in good (58.62%) than poor (20%) responders (*p* = 0.025). Similarly, proportions of patients carrying the synonymous SNP rs2070096 (*BARD1Thr351=*) (wv + vv) were higher in patients showing a pathological complete response (46.67%) than in those not showing this response (15.15%) (*p* = 0.031). The SNPs rs1058808 (*HER2Ala1170Pro*) and rs2070096 (*BARD1Thr351=*) were identified here as potential biomarkers of a good response to anti-HER2 treatment.

## 1. Introduction

The human epidermal growth factor receptor 2 gene (*HER2* or *ERBB2*) encodes a transmembrane tyrosine kinase receptor protein. This gene is involved in the regulation of cell growth, differentiation, survival, and invasion of other tissues [1]. Overexpression of *HER2* occurs in 20 to 30% of all breast cancer (BC) patients, and its association with an aggressive phenotype, HER2-positive BC, has been well established [2]. Neoadjuvant therapy, initially introduced for patients with inoperable breast cancer, is today an excellent choice for breast-conserving surgery in early stage breast tumors. This treatment targets a pathological complete response (pCR) detected upon surgery and has been found to improve disease-free survival and overall survival in patients with early stage HER2-positive BC [3].

The HER2 protein has an extracellular domain, which is the therapeutic target of monoclonal antibodies such as trastuzumab and pertuzumab. Trastuzumab (TZ) inhibits ligand-independent HER2 and HER3 signaling through the antibody-dependent cellular cytotoxicity-inducing PI3K/AKT pathway [4]. Pertuzumab (PZ) blocks HER2 dimerization and inhibits classic HER2-mediated cell-signaling cascades. The addition of TZ to chemotherapy (CT) nearly two decades ago improved disease outcomes in HER2-positive BC patients. For instance, a 2012 meta-analysis found that treatment regimens containing TZ resulted in significantly improved overall survival and disease-free survival (DFS) in early BC patients [5]. Neoadjuvant chemotherapy consisting of dual HER2 blockade (TZ plus PZ) increases the pCR rate in around 50–70% of patients, compared with TZ alone [6]. Recently, an analysis of 1763 patient-level data from five randomized trials designed to assess event-free survival in response to TZ, PZ, or both, as part of systemic neoadjuvant and adjuvant therapy for HER2-positive early BC, revealed a lower risk of breast cancer recurrence [7].

As neoadjuvant treatment, anti-HER2 drugs have shown different responses among HER2-positive BC patients. In a multicenter retrospective observational study in these patients treated with neoadjuvant CT plus TZ, failure to achieve pCR was associated with a significantly worse DFS compared to the rate detected in those showing a pCR [8]. While the addition of TZ significantly improved disease-free and overall survival, TZ resistance observed in some patients is a major clinical problem that remains poorly understood [9]. Several polymorphisms have been identified that show a relationship with the response of BC patients to anti-HER2 treatments. For example, Furrer et al. concluded that the rs1136201 SNP in the *HER2* gene could affect the TZ response, such that the presence of Val instead of Ile was associated with worse DFS [10]. Single nucleotide polymorphisms affecting the genes *BARD1* and *HER3* have been identified as potential biomarkers of a worse response to TZ-based treatment through abnormal cell signaling [11]. In effect, there is an urgent need to find genetic biomarkers that will help distinguish between HER2-positive BC patients who are more likely or not to respond well to anti-HER2 treatment.

The aim of this study was to assess possible associations between the efficacy of neoadjuvant anti-HER2 treatment in early HER2-positive BC patients and the presence of 28 SNPs affecting the 17 genes or DNA regions: *PON1, CAT, GSTP1*, *FCGR3*, *ATM*, *PIK3CA*, *HER3*, *BARD1*, *LDB2*, *BRINP1*, chr6 intergenic region, *RAB22A*, *TRPC6*, *LINC01060*, *EGFR*, *ABCB1*, and *HER2*. These polymorphisms were chosen because of either their known impacts on different cell processes (e.g., PI3K/AKT pathway, DNA damage repair, immune response regulation) or of evidence of a relationship with TZ/PZ treatment in BC patients.

## 2. Materials and Methods

### 2.1. Design and Patients

Fifty female patients diagnosed with HER2-positive breast cancer were treated at the Oncology Department of the University Hospital of Fuenlabrada (Madrid, Spain) over the period 2010 to 2018. Patients received TZ or a PZ/TZ combo and CT as neoadjuvant therapy for a pathological complete response and, thus, breast-conserving surgery. The following therapeutic regimens were used on the basis of current standard of care in each case: (i) anthracycline-based therapy (epirubicin, cyclophosphamide with weekly paclitaxel), (ii) carboplatin-docetaxel, or (iii) taxane monotherapy (weekly paclitaxel). Previously, HER2, ER (estrogen receptor), and PR (progesterone receptor) status was determined as part of the routine diagnostic procedure in each patient. Oncologists at the Oncology Department collected patient data and samples in accordance with the Declaration of Helsinki. The study protocol was approved by the Hospital’s Ethics Committee (identification code: APR 18/20, September 2018). Informed consent for the experimental use of biopsy specimens collected for diagnosis was provided by all patients.

To assess the response to treatment, the Miller–Payne tumor grading system was used: 1 = no tumor reduction, 2 = up to 30% tumor reduction, 3 = 30%–90% tumor reduction, 4 = >90% tumor reduction (considered close to pCR), and 5 = no invasive malignant cells identifiable in sections from the tumor site (considered pCR). The tumor response was assessed after 6 months of treatment at the time of surgical resection classifying patients as good responders (Miller–Payne grades 4 and 5) or poor responders (Miller–Payne grades 1, 2, and 3).

Overall survival (OS) was defined as the time from treatment onset to the date of any-cause death or last follow-up.

### 2.2. SNP Selection and Analysis

The 28 SNPs examined here were selected on the basis of reported evidence of their role as biomarkers in similar patient cohorts, or their key roles in cell processes related to the HER2 pathway i.e., [11,12,13]. These SNPs (DNA region or gene locations in brackets) were rs662, rs854560 (*PON1*); rs1001179 (*CAT*); rs1695 (*GSTP1*); rs396991 (*FCGR3*); rs11212617 (*ATM*); rs104886003, rs121913279, rs121913273 (*PIK3CA*); rs2229046, rs773123 (*HER3*); rs2070096 (*BARD1*); rs55756123 (*LDB2*); rs62568637 (*BRINP1*); rs4305714 (chr6 intergenic region); rs707557 (*RAB22A*); rs77679196 (*TRPC6*); rs7698718 (*LINC01060*); rs2293347, rs1140475 (*EGFR*); rs1045642 (*ABCB1*); and rs104886003, rs1136201, rs121913471, rs1057519738, rs1057519816, rs1057519862, rs121913470 (*HER2*).

Total DNA was extracted from 2.5 mm^3^ of paraffin-embedded tumor biopsies as required by the QIAamp^®^ DNA FFPE Tissue Kit (Qiagen, Germantown, MD, USA). We only included tumor specimens showing >50% cellularity. The DNA samples were re-suspended in DNAse-free water (50 µL). Concentrations and quality of DNA samples were quantified with the Nanodrop 2000 (Thermo Fisher Scientific, Wilmington, DE, USA). Genotyping of 28 SNPs was performed by the Spanish National Genotyping Center (CeGen-PRB2-ISCIII, http://www.usc.es/cegen/ accessed on 15 December 2022). The platforms iPlex^®^Gold chemistry and MassARRAY were used to analyze samples according to the manufacturer’s instructions (Agena Bioscience, San Diego, CA, USA). Assays were designed on GRCh38 version using Agena Bioscience MassARRAY Assay 4.0 software (Agena Bioscience, San Diego, CA, USA). All assays were performed in 384-well plates, including negative controls and a trio of Coriell samples (Na10861, Na11994 and Na11995) for quality control. Internal controls showed 100% reproducibility and genotyping success. All reactions were performed in duplicate.

### 2.3. Statistical Analysis

Baseline clinical characteristics are described for all participants. Quantitative variables are provided as medians with their interquartile range (IQR) or as the mean ± standard deviation (SD), according to their distribution (Shapiro–Wilk test for normality). For qualitative variables, absolute and relative frequencies are given in percentages. For each SNP, the association between genotype (w = wild-type allele; v = variant allele) and tumor response to the anti-HER2 treatment received (good or poor response) was determined using contingency tables and tested by the chi-squared test or Fisher’s exact test, as appropriate. Time-to-event data were analyzed using the Kaplan–Meier method. All calculations were performed using the Program Stata v.14.2 (StataCorp LLC, TX, USA). Significance was set at *p* ≤ 0.05.

## 3. Results

### 3.1. Patient Characteristics

The study cohort was comprised of 50 patients with breast cancer. Median age was 51.1 years (range 28.4–78.2 years), and all participants were female (100%, Table 1). Tumor locations were 38% right side and 62% left side. In 16%, 52%, and 32% of women, respectively, tumors were histology grades 1, 2, and 3. Ninety eight percent of participants had no metastasis. Only one patient presented mediastinal lymph node involvement upon diagnosis. While this was considered metastatic disease, the patient was included as the chemotherapy regimen was similar and the tumor was considered resectable if a response to therapy was produced. All patients were positive for HER2, ER, and PR. Most tumors were ductal (94%); the remaining 6% were lobular (Table 1).

The neoadjuvant anti-HER2 treatment received was TZ in 68% and TZ + PZ combo in the remaining 32%. Chemotherapy regimens based on current standard guidelines were carboplatin–docetaxel in 62% of patients, anthracycline-based therapy in 22%, and taxane monotherapy in 16%. After six months of therapy with TZ or PZ/TZ and CT, 30% of patients showed a pCR (Miller–Payne grade 5). A total of 66% of patients (33/50, Table 1) achieved a good response (>90% tumor reduction, grade 4 + 5), whereas 34% were considered poor responders as their tumors were reduced by <90% after six months of treatment (Table 2). When patients were stratified according to the anti-HER2 treatment received, TZ or TZ/PZ combo, rates of good responders were similar in both groups, 61.9% versus 75%, respectively (*p* = 0.614).

Allele and genotype frequencies for 28 SNPs affecting 17 genes (see Table 2) were estimated by direct counting. Allele frequencies were in the ranges shown by European populations available from IGSR (The International Genome Sample Resource) [14]. All the gene frequencies showed good agreement with Hardy–Weinberg equilibrium.

No association was detected between response to therapy and the clinical characteristics of the patients (data not shown). Nor was a link detected between conventional chemotherapy type and response.

### 3.2. Gene Variants in Relation to Treatment Response and Survival

Possible associations between the presence of a given allelic variant and treatment response were analyzed for the 28 SNPs included in this study. Five of them were excluded from analysis as they showed no genetic variability in that 100% of the patients were homozygous for the wild-type allele: one in the *PIK3CA* gene (rs121913273) and four more in the *HER2* gene (rs1057519738, rs1057519816, rs1057519862, rs121913470; Table 2).

When patients were divided into two categories of genotypes, with (wv + vv) or without the mutation (ww), no SNP differed significantly in terms of genotype rates between good and poor responders. However, when patients were grouped by the presence or not of a double mutation, homozygous vv versus wv + ww, the proportions of patients with a double mutation for rs1058808 in the gene *HER2* was significantly higher in the good responders (58.62%) than poor responders (20%) (*p* = 0.025, Table 3). This SNP, in which alanine is replaced by proline at amino acid residue 1170, thus seems to be associated with a better response to anti-HER2 treatment (TZ and TZ + PZ combo) when a patient is homozygous for the allelic variant. When participants were stratified according to the drug prescribed (34 patients treated with TZ alone versus 16 patients receiving the TZ + PZ combo), only in the TZ group was a significantly higher proportion of a double mutation homozygous for the same SNP rs1058808 observed in good responders (57.89%) compared to poor responders (18.18%) (*p* = 0.034). This difference was not observed in the TZ + PZ combo group.

We also examined a possible link between the studied SNPs and achieving a pCR (Miller–Payne grade 5) in patients homozygous for the variant allele (vv) versus wv + ww with no significant result for any SNP. However, patients grouped into those homozygous and heterozygous for the variant allele (vv + wv) showed a significant difference only in the case of rs2070096 (*BARD1*) in that the proportion of patients with the mutation was significantly higher among those showing a pCR (46.67%) than those not showing this response (15.15%) (*p* = 0.031, Table 4). Seven SNPs were excluded from this analysis because of the lack of genotype variability needed for this comparison. When subgroups of patients prescribed different drugs (TZ or TZ + PZ) were separately analyzed, the proportion of patients with the mutation (wv + vv) for rs2070096 in the TZ/PZ subgroup was significantly higher in those showing a pCR (66.67%) compared to those not showing this response (0%) (*p* = 0.011).

As only four patients died at 50 to 65 months (five-year survival was 87.02%; 95% CI: 68.98–95.95), it was not possible to assess the relationship between each SNP and survival.

## 4. Discussion

Neoadjuvant therapy based on anti-HER2 drugs is a good option for breast-conserving surgery in patients with early breast cancer as it gives rise to a very good pCR at the time of surgery. Consistently, anti-HER2 treatments have been described to improve survival and lead to a lower risk of recurrence in early HER2-positive BC patients [3,7]. However, mechanisms of resistance to this type of treatment are so far poorly understood, and there is thus a need to identify genetic biomarkers of susceptibility or resistance to provide more efficient and personalized treatment.

Here, we characterized 28 SNPs in 17 genes, chosen because of their known implications for several cell mechanisms related to cancer progression. Among them, a significant relationship was noted between being homozygous for the *HER2Ala1170Pro* (rs1058808) mutation and showing a good response to treatment both in the whole sample of patients (*p* = 0.025, Table 3) and in the subgroup of patients receiving TZ alone as neoadjuvant therapy (*p* = 0.034). This significant association was not observed in the group TZ + PZ combo, in which a good response was shown by six of seven women with the variant homozygous *HER2Ala1170Pro* genotype (Pro/Pro). The small number of samples in the TZ + PZ combo group (only 16) could be a limiting factor of our study. This common SNP, in which alanine is replaced by proline at residue 1170, is frequent in European populations (Table 2), although its functional significance remains unknown. Other authors have also related the proline allele to a higher frequency of *HER2* overexpression in breast tumors [15], along with a tendency for loss of the wild-type allele at codon 1170 during carcinogenesis. In effect, the frequency of the Pro allele is significantly higher in patients with cancer than in healthy subjects and has been linked to a worse prognosis [16]. The relationship between the presence of SNPs in the *HER2* gene and the response to anti-HER2 treatment has been less explored. Our results indicate than the Pro/Pro genotype is associated with a good response to anti-HER2 drugs (>90% tumor reduction): in the whole patient sample, 58.62% of good responders were of the Pro/Pro genotype versus only 20% for the remaining genotypes (*p* = 0.025, Table 3). The same significant associations were observed in the subgroup of patients given TZ monotherapy (*n* = 34): 57.89% of Pro/Pro were good responders versus 18.18% of the remaining genotypes (*p* = 0.34, data not shown in the tables). However, this difference was not detected in the TZ + PZ subgroup comprising a small number of patients (*n* = 16). Furrer et al. found no association between the *HER2Ala1170Pro* SNP and DFS in a cohort of 237 women with early HER2-positive BC treated with TZ [10]. Stanton et al. observed a non-significant trend associating the Pro/Pro genotype with TZ cardiomyopathy, suggesting black ethnicity as a possible contributing factor [17]. Owing to our small sample size, we were unable to explore these interesting factors that warrant further investigation in larger studies. In fact, several authors have reported the considerable therapeutic benefits of TZ treatment in HER2-positive BC despite its association with cardiotoxicity and a critical need to elucidate the molecular mechanisms of this cardiotoxicity, e.g., [18]. Although few studies have addressed the role of the *HER2Ala1170Pro* SNP, the impact of this polymorphism on BC treatment with TZ seems relevant. There is thus a need to clarify both its efficacy and cardiotoxicity so that patients who might benefit from this treatment with less risk can be identified.

We found no link between the other *HER2* SNP rs1136201 (*HER2Ile655Val* polymorphism) and the response shown to treatment (0.461, Table 3). Notwithstanding, interesting albeit controversial results have been described by others for this polymorphism in terms of patient survival. Furrer et al. observed a significantly worse DFS in Val/Val patients versus the rest of genotypes during TZ treatment [10], while Han et al. reported significantly better DFS in patients with the Ile/Val or Val/Val genotypes [19]. We were unable to examine the association between the studied SNPs and survival due to the low mortality observed (five-year survival 87.02%; 95% CI: 68.98–95.95). Similarly, we obtained no significant results for rs773123 in the *HER3* gene. In contrast, Coté et al. observed that women with a heterozygous genotype for this SNP were significantly more likely to relapse in response to a TZ + CT regimen than those not receiving TZ [11]. Consequently, both SNPs in the *HER2* and *HER3* genes seem to be associated in some way with the efficacy of anti-HER2 treatments, warranting the assessment of this issue in larger studies.

The addition of TZ to neoadjuvant treatment or dual HER2 blockade (TZ + PZ combo) is known to significantly increase pCR rates in early BC patient cohorts, e.g., [20]. In our study, 15 out of a total of 50 patients (30%) showed a pCR (Miller–Payne grade = 5). Neoadjuvant chemotherapy with TZ + PZ combo was found to increase the pCR rate compared with TZ alone to around 50–70% [6]. We also obtained a better pCR rate in our TZ + PZ (6/16, 37.50%) than TZ subgroup (9/34, 26.47%), but the difference lacked significance. When we examined the 28 selected SNPs, pCR was significantly associated with the allelic variant rs2070096 (*BARD1Thr351=*). Accordingly, the proportion of patients with the variant allele (wv + vv) was significantly higher among those showing a pCR (46.67%) than those not showing a pCR (15.15%) (*p* = 0.031, Table 4). This significant association was also detected in the subgroup of patients given the combo treatment (66.67% vs. 0%, respectively, *p* = 0.011).

BARD1 (BRCA1-associated RING domain 1) is an essential gene related to breast cancer development that encodes a protein that interacts with the N-terminal region of BRCA1. Mutations in the tumor suppressor gene *BRCA1* or in any related gene are the most common cause of homologous recombination deficiency [21]. Although this rs2070096 SNP in the *BARD1* gene consists of a C > G substitution in the DNA, its consequence is the protein Thr351Thr. Until recently, this type of synonymous variant was believed to be silent because of its little to null impact on the ensuing protein. However, it is generally accepted that codon bias contributes to translation efficiency by tuning the elongation rate of the process [22]. Several studies have shown that synonymous SNPs could play an important role in the functionality of the cancer cell and in the response of patients to targeted therapies as the resulting aberrant mRNA splicing or mRNA instability could affect protein conformation with clinical consequences [23]. Here, we identified for the first time an association between the variant allele of rs2070096 SNP in the *BARD1* gene and a Miller–Payne tumor grade 5, that is, a pCR, not only in the whole patient cohort but also in the subgroup of 16 patients receiving TZ/PZ combo treatment. In contrast, Coté et al. found that patients heterozygous for the BARD1 rs2070096 SNP were more likely to relapse when on a TZ + CT-based treatment compared to a non-TZ + CT-based treatment [11]. It should be mentioned that we did not compare a TZ + CT and a non-TZ + CT treatment group, so our results are not really commensurable. These authors proposed that impairment of the correct homologous recombination deficiency pathway, with the possible involvement of variant RB1, would result in altered sensitivity to anti-HER2 treatments. Unfortunately, as far as we know, no other study has examined the possible link between this relevant *BARD1Thr351=* SNP and a modified efficacy of TZ + CT treatment. Indeed, this type of study would be useful to clarify the role of this rs2070096 SNP in anti-HER2 treatment in early BC patients.

The identification of genetic biomarkers to optimize personalized treatments is the focus of all these pharmacogenetic studies. Approximately one-third of patients with early HER2-positive BC treated with TZ show cancer relapse [24]. This highlights a need to find optimal biomarkers of the efficacy of anti-HER2 treatment for this aggressive BC phenotype. Previous studies by our group have suggested the potential utility of CNV polymorphisms because of the association between *FGFR1* gene amplification and a poor response to anti-HER2 treatments [25]. It is probably the case that genes other than *FGFR1* and *HER2*, with roles in the metabolic pathway PI3K/AKT and with known consequences on cell cycle regulation via inhibition of p53, would be interesting candidates as biomarkers. Specific miRNAs have also been associated with resistance to anti-HER2 therapies. Hence, miR-200b, miR-135b, and miR-29a have been identified as upregulated and miR-224 as downregulated in trastuzumab-resistant serums from HER2-positive BC patients [26]. Our study does have limitations that must be taken into consideration such as a small sample size and incomplete clinicopathological information in some cases. Nonetheless, we here propose two SNPs, rs1058808 (*HER2Ala1170Pro*) and rs2070096 (*BARD1Thr351=*), as potential biomarkers of a good response to neoadjuvant treatment with trastuzumab and pertuzumab for early BC. These results provide direction for future studies designed to identify biomarkers for more personalized treatments.

## 5. Conclusions

The SNPs rs1058808 (*HER2Ala1170Pro*) and rs2070096 (*BARD1Thr351=*) could be considered as potential biomarkers of a good response to neoadjuvant treatment with trastuzumab and pertuzumab for early BC patients.

## Figures and Tables

**Table 1 cancers-15-00763-t001:** Baseline characteristics of the 50 patients enrolled in this study.

Patient Characteristics	No. (%)
Median age (years)	51.1, range 28.4–78.2
Gender - Female - Male	50 (100) --
Tumor location - Right side - Left side	19 (38) 31 (62)
Histology type - Ductal - Lobular	47 (94) 3 (6)
Histology grade - Grade 1 - Grade 2 - Grade 3	8 (16) 26 (52) 16 (32)
HER2 status - Positive - Negative	50 (100) --
ER status - Positive - Negative	50 (100) --
PR status - Positive - Negative	50 (100) --
Metastasis - No - Yes	49 (98) 1 (2)
Miller–Payne response grade - 1 - 2 - 3 - 4 - 5	1 (2) 8 (16) 8 (16) 18 (36) 15 (30)
Anti-HER2 drug - Trastuzumab - Trastuzumab + pertuzumab	34 (68) 16 (32)
Chemotherapy - Anthracycline-based therapy - Carboplatin-docetaxel therapy - Taxane monotherapy	11 (22) 31 (62) 8 (16)

**Table 2 cancers-15-00763-t002:** Genotypes shown as allelic frequencies of the 28 SNPs examined (w = wild-type allele, v = variant allele). Genes are classified according to gene ontology (the biological processes they affect). Allele frequencies from the 1000 Genomes database are indicated (-- = no data available).

Gene Ontology	SNP Gene or Region	Genotypes ww wv vv	Patients		Allele Frequencies w v	1000 Genomes Allele Frequency (Europeans)
			*n*	%		
Xenobiotic metabolism	rs1045642 A > G *ABCB1*	AA	10	20.00	A = 0.46 G = 0.54	A = 0.52 G = 0.48
		AG	26	52.00		
		GG	14	28.00		
DNA damage check-point hypoxia	rs11212617 C > A *ATM*	CC	10	20.83	C = 0.44 A = 0.56	C = 0.38 A = 0.62
		CA	22	45.83		
		AA	16	33.33		
DNA repair/polyubiquitination	rs2070096 C > G *BARD1*	CC	36	75.00	C = 0.85 G = 0.15	C = 0.82 G = 0.18
		CG	10	20.83		
		GG	2	4.17		
Cell death/cell cycle	rs62568637 G > A *BRINP1*	GG	47	95.92	G = 0.98 A = 0.02	G = 0.98 A = 0.02
		GA	2	4.08		
		AA	0	0.00		
Hypoxia/response to ROS (reactive oxygen species)	rs1001179 C > T *CAT*	CC	27	62.79	C = 0.77 T = 0.23	C = 0.77 T = 0.23
		CT	12	27.91		
		TT	4	9.30		
MAPK cascade/protein phosphorylation	rs2293347 C > T *EGFR*	CC	42	84.00	C = 0.92 T = 0.08	C = 0.89 T = 0.11
		CT	8	16.00		
		TT	0	0.00		
	rs1140475 T > C *EGFR*	TT	0	0.00	T = 0.12 C = 0.88	T = 0.11 C = 0.89
		TC	12	24.49		
		CC	37	75.51		
Immune response regulation	rs396991 A > C *FCGR3*	AA	17	34.69	A = 0.60 C = 0.40	A = 0.66 C = 0.34
		AC	25	51.02		
		CC	7	14.29		
Lipid metabolism	rs1695 A > G *GSTP1*	AA	24	51.06	A = 0.76 G = 0.24	A = 0.67 G = 0.33
		AG	23	48.94		
		GG	0	0.00		
Signal transduction/protein phosphorylation Signal transduction/protein phosphorylation	rs1057519738 G > A *HER2*	GG	49	100.00	G = 1.00 A = 0.00	G = 1.00 A = 0.00
		GA	0	0.00		
		AA	0	0.00		
	rs1057519816 C > A,T *HER2*	CC	50	100.00	C = 1.00 A = 0.00	G = 1.00 A = 0.00
		CA	0	0.00		
		AA	0	0.00		
	rs1057519862 G > A *HER2*	GG	50	100.00	G = 1.00 A = 0.00	G = 1.00 A = 0.00
		GA	0	0.00		
		AA	0	0.00		
	rs121913470 T > C,G *HER2*	TT	49	100.00	T = 1.00 C = 0.00	--
		TC	0	0.00		
		CC	0	0.00		
	rs121913471 G > T *HER2*	GG	48	97.96	G = 0.99 T = 0.01	--
		GT	1	2.04		
		TT	0	0.00		
	rs1058808 C > G *HER2*	CC	11	25.00	C = 0.40 G = 0.60	C = 0.33 G = 0.67
		CG	13	29.55		
		GG	20	45.45		
	rs1136201 A > G *HER2*	AA	35	70.00	A = 0.75 G = 0.25	A = 0.75 G = 0.25
		AG	5	10.00		
		GG	10	20.00		
Signaling pathway/ tyrosine kinase	rs2229046 T > C *HER3*	TT	41	85.42	T = 0.93 C = 0.07	T = 0.93 C = 0.07
		TC	7	14.58		
		CC	0	0.00		
	rs773123 A > T *HER3*	AA	41	83.67	A = 0.91 T = 0.09	A = 0.89 T = 0.11
		AT	7	14.29		
		TT	1	2.04		
RNA polymerase II transcription regulation	rs55756123 C > T *LDB2*	CC	48	97.96	C = 0.99 T = 0.01	C = 0.99 T = 0.01
		CT	1	2.04		
		TT	0	0.00		
Angiogenesis	rs104886003 G > A *PIK3CA*	GG	47	95.92	G = 0.98 A = 0.02	G = 1.00 A = 0.00
		GA	2	4.08		
		AA	0	0.00		
	rs121913279 A > G *PIK3CA*	AA	46	92.00	A = 0.96 G = 0.04	A = 1.00 G = 0.00
		AG	4	8.00		
		GG	0	0.00		
	rs121913273 G > C,A *PIK3CA*	GG	50	100.00	G = 1.00 C = 0.00	G = 1.00 C = 0.00
		GC	0	0.00		
		CC	0	0.00		
Lipid metabolism process	rs662 T > C *PON1*	TT	23	46.00	T = 0.67 C = 0.33	T = 0.71 C = 0.29
		TC	21	42.00		
		CC	6	12.00		
	rs854560 A > T *PON1*	AA	19	40.43	A = 0.64 T = 0.36	A = 0.64 T = 0.36
		AT	22	46.81		
		TT	6	12.77		
Endocytosis	rs707557 C > T *RAB22A*	CC	21	42.00	C = 0.68 T = 0.32	C = 0.59 T = 0.41
		CT	26	52.00		
		TT	3	6.00		
Ion transport	rs77679196 G > A,C *TRPC6*	GG	48	96.00	G = 0.98 A = 0.02	G = 0.99 A = 0.01 C = 0–0.00003
		GA	2	4.00		
		AA	0	0.00		
Not described	rs4305714 C > T chr6 intergenic region	CC	24	48.98	C = 0.71 T = 0.29	C = 0.78 T = 0.22
		CT	22	44.90		
		TT	3	6.12		
	rs7698718 C > A *LINC01060*	CC	20	57.14	C = 0.76 A = 0.24	C = 0.83 A = 0.17

**Table 3 cancers-15-00763-t003:** Genotype frequencies (homozygous mutation versus the rest) in good responders and poor responders for 14 SNPs. *p*-value corresponds to Fisher’s exact test. Significant value is highlighted in bold. w = wild-type allele and v = variant allele.

SNP (Gene or Region)	Homozygous Mutation (No = ww + wv Yes = vv)	Patients		Good Responders		Poor Responders		*p*-Value
		*n*	%	*n*	%	*n*	%	
rs1045642 *(ABCB1)*	No	36	72.00	25	75.76	11	64.71	0.511
	Yes	14	28.00	8	24.24	6	35.29	
rs11212617 (*ATM*)	No	32	66.67	21	67.74	11	64.71	1.000
	Yes	16	33.33	10	32.26	6	35.29	
rs2070096 (*BARD1)*	No	46	95.83	31	96.88	15	93.75	1.000
	Yes	2	4.17	1	3.13	1	6.25	
rs1001179 (*CAT)*	No	39	90.70	26	92.86	13	86.77	0.602
	Yes	4	9.30	2	7.14	2	13.33	
rs1140475 (*EGFR*)	No	12	24.49	7	21.88	5	29.41	0.729
	Yes	37	75.51	25	78.13	12	70.59	
rs396991 (*FCGR3*)	No	42	85.71	28	87.5	14	82.35	0.681
	Yes	7	14.29	4	12.5	3	17.65	
rs1058808 (*HER2*)	No	24	54.55	12	41.38	12	80	**0.025**
	Yes	20	45.45	17	58.62	3	20	
rs1136201 (*HER2*)	No	40	80	25	75.76	15	88.24	0.461
	Yes	10	20	8	24.24	2	11.76	
rs773123 (*HER3*)	No	48	97.96	33	100	15	93.75	0.327
	Yes	1	2.04			1	6.25	
rs4305714 (chr6 intergenic region)	No	46	93.88	30	93.75	16	94.12	1.000
	Yes	3	6.12	2	6.25	1	5.88	
rs7698718 (*LINC01060*)	No	33	94.29	18	90.00	15	100.00	0.496
	Yes	2	5.71	2	10.00			
rs854560 (*PON1)*	No	41	87.23	26	81.25	15	100.00	0.157
	Yes	6	12.77	6	18.75			
rs662 (*PON1)*	No	44	88.00	29	87.88	15	88.24	1.000
	Yes	6	12.00	4	12.12	2	11.76	
rs707557 (*RAB22A*)	No	33	94.00	32	96.97	15	88.24	1.00
	Yes	2	6.00	1	3.03	2	11.76	

**Table 4 cancers-15-00763-t004:** Genotype frequencies (presence of mutation or not) in patients showing pCR versus those not showing pCR for 21 SNPs. *p*-value corresponds to Fisher’s exact test. Significant value is highlighted in bold. w = wild-type allele and v = variant allele.

SNP (Gene or Region)	Presence of Mutation (No = ww Yes = wv + vv)	Patients		No pCR		pCR		*p*-Value
		*n*	%	*n*	%	*n*	%	
rs1045642 (*ABCB1*)	ww	10	20	7	20	3	20	1.000
	wv + vv	40	80	28	80	12	80	
rs11212617 (*ATM*)	ww	10	20.41	8	22.86	2	14.29	0.702
	wv + vv	39	79.59	27	77.14	12	85.71	
rs2070096 (*BARD1*)	ww	36	75	28	84.85	8	53.33	**0.031**
	wv + vv	12	25	5	15.15	7	46.67	
rs62568637 (*BRINP1*)	ww	47	95.92	34	97.14	13	92.86	0.494
	wv + vv	2	4.08	1	2.86	1	7.14	
rs1001179 *(CAT)*	ww	27	62.79	19	63.33	8	61.54	1.000
	wv + vv	16	37.21	11	36.67	5	38.46	
rs2293347 (*EGFR*)	ww	42	84	29	82.86	13	86.67	1.000
	wv + vv	8	16	6	17.14	2	13.33	
rs396991 (*FCGR3*)	ww	17	34.69	13	37.14	10	71.43	0.743
	wv + vv	32	65.31	22	62.86	8	53.33	
rs1695 (*GSTP1*)	ww	24	51.06	14	43.75	10	66.67	0.212
	wv + vv	23	48.94	18	56.25	5	33.33	
rs121913471 (*HER2*)	ww	48	97.96	34	97.14	14	100	1.000
	wv + vv	1	2.04	1	2.86			
rs1058808 (*HER2*)	ww	11	25	6	20	5	35.71	0.287
	wv + vv	33	75	24	80	9	64.29	
rs1136201 (*HER2*)	ww	35	70	24	68.57	11	73.33	1.000
	wv + vv	15	30	11	31.43	4	26.67	
rs2229046 (*HER3*)	ww	41	85.42	28	82.35	13	92.86	0.656
	wv + vv	7	14.58	6	17.65	1	7.14	
rs773123 (*HER3*)	ww	41	83.67	28	82.35	13	86.67	1.000
	wv + vv	8	16.33	6	17.65	2	13.33	
rs7698718 (*LINC01060*)	ww	20	57.14	15	60	5	50	0.712
	wv + vv	15	42.86	10	40	5	50	
rs55756123 (*LDB2*)	ww	48	97.96	34	100	14	93.33	0.306
	wv + vv	1	2.04			1	6.67	
rs104886003 (*PIK3CA*)	ww	47	95.92	33	94.29	14	100	1.00
	wv + vv	2	4.08	2	5.71			
rs121913279 (*PIK3CA*)	ww	46	92	32	91.43	14	93.33	1.00
	wv + vv	4	8	3	8.57	1	6.67	
rs662 (*PON1*)	ww	23	46	18	51.43	5	33.33	0.355
	wv + vv	27	54	17	48.57	10	66.67	
rs707557 (*RAB22A*)	ww	21	42	15	42.86	6	40	1.000
	wv + vv	29	58	20	57.14	9	60	
rs77679196 (*TRPC6*)	ww	48	96	34	97.14	14	93.33	0.514
	wv + vv	2	4	1	2.86	1	6.67	
rs4305714 (chr6 intergenic region)	ww	24	48.98	16	47.06	8	53.33	0.762
	wv + vv	25	51.02	18	52.94	7	46.67	

## Data Availability

Not applicable.

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
