# Peer review of "HER2 and BARD1 Polymorphisms in Early HER2-Positive Breast Cancer Patients: Relationship with Response to Neoadjuvant Anti-HER2 Treatment"

_cancers, 2023, doi:10.3390/cancers15030763_

Round 1
Reviewer 1 Report
The present study aimed to identify SNPs associated with the response to antiHER2 neoadjuvant therapy in HER2+ ER+ PR+ patients. This is an important undertaking, which may lead to improved patient selection and prevention of relapse. However, the study is limited by small numbers of patients and conflicting literature is not always addressed. I also have comments about missing controls and the methodology used, as described below.
- Line 61 – reference [4] is not a suitable citation of ADCC as the primary mode of action of trastuzumab.
- Line 166 – the reference to Table 2 is wrong, the text refers to Table 1.
- In the text, the authors describe the absence of metastatic patients at diagnosis in their cohort (lines 156-8). However, Table 1 shows that there was one metastatic patient at baseline. Could the authors explain this apparent discrepancy?
- It is not clear in Materials and Methods whether the presence or absence of SNPs was determined using the diagnostic or the surgical specimens. This has implications for the analysis of the correlation between SNPs and antiHER2 efficacy.
- It is not clear why the authors focussed on endocrine breast cancer. Could the statistical power of the study be improved by adding non-endocrine HER2+ patients ?
- The authors cited previous studies (Furrer et al and Cote et al), which previously showed an association between increase likelihood of a poor response to antiHER2 and the presence of rs1136201 (HER2), rs2070096 (BARD1) and rs773123 (HER3) SNPs. However, the data in Table 3 shows no association between response to antiHER2 and the presence of these SNPs. Similarly, in Table 4, there is no association between the presence of rs1136201 (HER2) and rs773123 (HER3) and pCR. In fact, the presence of rs2070096 (BARD1) is associated with a good response (pCR), the opposite results from Cote et al. While rs1136201 (HER2) is discussed at lines 253-60, the present manuscript would be improved by discussing the discrepancies between the findings of their studies and the literature pertaining to rs2070096 (BARD1) and rs773123 (HER3).
- Please provide the break-down of all Miller-Payne scores for each grade 1, 2, 3, 4 and 5 in Table 1.
- In tables 2,3 and4, the first rows contain information in bold. Is there a missing key in legend or is it mis-formatting ?
- Why is homozygous rs1058808 (HER2) only associated with a good response to TZ and not TZ/PZ. If rs1058808 (HER2) is a biomarker of a favourable response to antiHER2, it should predict efficacy of TZ and also of TZ/PZ. The manuscript would be improved by discussing this point.
- The analysis of SNPs was performed using “paraffin-embedded tumor biopsies” (line 128). Breast tumour may be strongly stromal. What is the cellularity of the samples used?
If the SNPs come from the stromal compartment of the tumour, then these SNPs are probably of germline origin. Are the SNPs found in this study germline or tumour-specific variants? This has implications for biomarker testing.
- Is there a known association between the SNPs tested in this study and the response to conventional chemotherapy (anthracyclin, platinums, taxanes) in breast cancer? The study methodology does not control for the fact that all patients received different chemotherapy regimens. The association between SNPs and responders to antiHER2 could be influenced by the chemotherapy regimen. Could the analysis of SNPs as predictive biomarker of antiHER2 response be performed within each of the 3 chemotherapy groups?
- The authors found significant associations between SNPs and response to antiHER2 using the comparison vv vs. wv+vv (Table 3), but when comparing ww vs. wv+vv. However, the authors only found significant associations between SNPs and pCR using the comparison ww vs. wv+vv (Table 3). Could the authors also comment on the results of ww vs. vv both for both the response to antiHER2 and for pCR ?
- The authors concluded that HER2Ala1170Pro (rs1058808) may be used as a predictive biomarker of antiHER2 efficacy. However, previous work cited in discussion (ref 17, lines 250-52), shows that this SNP is also associated with a higher risk of cardiotoxicity. The discussion would be improved by taking this into account to re-assess the value of HER2Ala1170Pro (rs1058808) as a predictive biomarker of treatment efficacy.
Author Response
Dear Reviewer,
Thank you very much for your revision. Our manuscript has been improved with your comments and suggestions. Besides, English language and style have been checked again (attached file).
Sincerely,
Ana

Reviewer 2 Report
Thanks for inviting me to evaluate the article titled ‘HER2 and BARD1 polymorphisms in early HER2-positive 2 breast cancer patients: relationship with response to neoadjuvant anti-HER2 treatment’. In this paper, the authors aimed to assess possible associations between the efficacy of neoadjuvant anti-HER2 treatment in early HER2-positive BC patients and the presence of 28 88 SNPs affecting the 17 genes or DNA regions. The authors identified two SNPs rs1058808 (HER2Ala1170Pro) and rs2070096 41 (BARD1Thr351=) as potential biomarkers of a good response to antiHER2 treatment. The text is well-arranged, and the logic is clear. Also, the English writing are fine.
Author Response
Dear reviewer,
Thank you very much for your revision
English language and style have been checked again (attached file). Manuscript has been improved with suggestion and comments of two reviewers.
Sincerely,
Ana Fernández-Santander, PhD

Round 2
Reviewer 1 Report
Thank you for considering my comments. I believe the manuscript, in its current revised form, will be of interest to the readers of Cancers.